# Oligomeric Proanthocyanidins Ameliorate Cadmium-Induced Senescence of Osteocytes Through Combating Oxidative Stress and Inflammation

**DOI:** 10.3390/antiox13121515

**Published:** 2024-12-12

**Authors:** Gengsheng Yu, Zehao Wang, Anqing Gong, Xiaohui Fu, Naineng Chen, Dehui Zhou, Yawen Li, Zongping Liu, Xishuai Tong

**Affiliations:** 1Joint International Research Laboratory of Agriculture and Agri-Product Safety of the Ministry of Education of China, Institute of Agricultural Science and Technology Development, College of Veterinary Medicine, Yangzhou University, Yangzhou 225009, China; dz120220012@stu.yzu.edu.cn (G.Y.); 13096616986@163.com (Z.W.); mz120221680@stu.yzu.edu.cn (A.G.); mz120211509@stu.yzu.edu.cn (X.F.); mz120231654@stu.yzu.edu.cn (N.C.); zhoudehui0507@163.com (D.Z.); 15062797581@163.com (Y.L.); 2Jiangsu Co-Innovation Center for Prevention and Control of Important Animal Infectious Diseases and Zoonoses, Yangzhou 225009, China; 3Jiangsu Key Laboratory of Zoonosis, Yangzhou 225009, China; 4Donghai County Animal Husbandry and Veterinarian Station, Lianyungang 222399, China

**Keywords:** oligomeric proanthocyanidins (OPC), cadmium (Cd), osteocytes, senescence, oxidative stress, inflammation

## Abstract

Osteocyte senescence is associated with skeletal dysfunction, but how to prevent bone loss and find the effective therapeutic targets is a potential scientific concern. Cadmium (Cd) is a widespread environmental contaminant that causes substantial bone damage in both animals and humans. Oligomeric proanthocyanidins (OPC) are naturally polyphenolic substances found in various plants and demonstrate significant anti-senescence potential. Here, we investigated the protective effects of OPC against Cd-induced senescence of osteocytes and identify potential regulatory mechanisms. OPC alleviated Cd-induced senescence of osteocytes by attenuating cell cycle arrest, reducing ROS accumulation, and suppressing pro-inflammatory responses in vitro. Furthermore, OPC effectively prevented the Cd-induced breakdown of dendritic synapses in osteocytes in vitro. Correspondingly, OPC ameliorated Cd-induced damage of osteocytes through anti-senescence activity in vivo. Taken together, our results establish OPC as a promising therapeutic agent that ameliorates Cd-induced osteocyte senescence by mitigating oxidative stress and inflammatory responses.

## 1. Introduction

Cadmium (Cd) is a pervasive environmental contaminant that can accumulate in animal tissues through multiple exposure routes including water, food, and soil, thereby posing a serious threat to both animals and humans [1]. Moreover, the skeletal system represents one of the primary target organs of Cd toxicity; even low-level Cd exposure can affect bone health [2]. Within the bone matrix, osteocytes constitute the predominant cellular population, residing within the mineralized bone matrix and controlling bone remodeling by secreting signaling molecules [3,4]. Cd-induced osteotoxicity in both osteoclasts and osteoblasts has been extensively reported, characterized by enhanced osteoblast apoptosis and inhibited mineralization [5,6]. Likewise, both direct and indirect Cd exposure has been demonstrated to augment osteoclastic differentiation and bone resorption in murine models [7,8]. Our recent investigations revealed that Cd could impair bone growth in mice and disrupt autophagy in murine long bone osteocyte-Y4 (MLO-Y4) cells through blocking the PI3K/AKT/mTOR pathway in vitro [9]. Although it has shown that Cd has osteotoxicity in osteocytes, the specific mechanism remains incompletely understood. Therefore, further exploration is needed to reveal the osteotoxicity of Cd on osteocytes.

While skeletal aging is an inevitable biological process, environmental factors can accelerate the premature aging of the skeleton [10]. Cellular senescence is characterized by increased metabolic activity, chromatin modifications, and cellular growth arrest [11]. Within cells, various stressors activate senescence-inducing pathways, including the p16/p21/p53 pathways, which subsequently trigger senescence mediators such as NF-κB, IL-1, and IL-6, and these mediators orchestrate the expression of the senescence-associated secretory phenotype (SASP), a range of cytokines including pro-inflammatory factors, chemokines, and proteases [12]. Notably, senescent cells generate elevated levels of reactive oxygen species (ROS), which induces DNA damage, and mitochondrial dysfunction, thereby establishing a positive feedback loop of oxidative stress and cellular senescence [13,14]. Recent investigations have revealed that Cd exposure induces premature senescence in cells by mechanisms involving oxidative stress and inflammation [15]. Previous research has demonstrated that senescent osteocytes accumulate in aging bone and induce senescence in neighboring healthy cells by secreting the senescence-associated secretory phenotype (SASP) into the surrounding bone microenvironment [16]. According to a recent study, targeting senescent bone marrow adipocytes or the SASP attenuates glucocorticoid-induced bone loss [17]. Overall, keeping osteocytes alive and healthy may be a new strategy for maintaining bone health. However, whether Cd exposure causes osteocyte senescence is largely unknown, and the potential mechanisms need to be further clarified.

Oligomeric proanthocyanidins (OPC), a naturally occurring polyphenolic flavonoid molecule extracted from various botanical sources, exhibits pleiotropic pharmacological properties, including antioxidation [18], anti-inflammation [19], and anti-aging [20]. In vitro, proanthocyanidins from sea buckthorn extract significantly prevent hydrogen peroxide-induced senescence in human skin fibroblasts through eliminating excess ROS and boosting glutathione (GSH) and superoxide dismutase (SOD) activity [20]. In vivo, procyanidin C1 (OPCC1), a polyphenolic component of Grape seed extract (GSE), significantly extends murine lifespan through the suppression of SASP production and selectively removing senescent cells [21]. Recent investigations from our laboratory demonstrated that dietary supplementation with OPC improves skeletal development in laying chicks via controlling the differentiation and function of osteoblasts and osteoclasts [22]. A recent study suggests that resveratrol attenuates diabetic periodontitis-induced iron death in alveolar bone cells by exerting anti-inflammatory and antioxidant capacities [23], suggesting an important role for plant-derived polyphenols in improving bone cell health. However, the effects of OPC on osteocyte senescence and functionality remain largely unexplored.

The aim of this study was to investigate the effects of Cd exposure on osteocyte senescence and to explore the potential protective mechanisms of OPC. In vitro, real-time cell analysis (RTCA), Senescence β-Galactosidase staining (SA-β-gal staining), immunofluorescence, flow cytometry, ELISA analysis, scanning electron microscopy, and Western blot analysis showed that Cd exposure induced osteocyte senescence, characterized by decreased cell viability, cell cycle arrest, mitochondrial dysfunction, DNA damage, oxidative stress, and inflammatory response. In vivo, H&E staining, immunohistochemistry, and Western blot analysis showed that Cd exposure induced osteocyte senescence and a decrease in the number of osteocytes. Additionally, OPC significantly ameliorated Cd-induced damage by exerting antioxidant and anti-inflammatory abilities. This study provides important insights into the toxicological mechanisms of Cd-induced osteocyte senescence and the intervention potential of OPC.

## 2. Materials and Methods

### 2.1. Reagents and Antibodies

Cadmium chloride (purity ≥ 99.99%) was acquired from Sigma-Aldrich (St. Louis, MO, USA). OPC (purity ≥ 95%) were obtained from Shanghai Yuanye Biotechnology Co., Ltd. (Shanghai, China). α-MEM was obtained from Gibco (Waltham, MA, USA). The CCK-8 kit and Annexin V-FITC/Propidium Iodide (PI) kit were obtained from Yeasen Biotechnology Co., Ltd. (Shanghai, China). The SA-β-gal staining kit was obtained from Solarbio Science & Technology Co., Ltd. (Beijing, China). The DCFH-DA staining kit, cell cycle analysis kit and ATP assay kit were acquired from Beyotime Biotechnology Inc. (Shanghai, China). The kFlour488 Click-iT EdU Kit was purchased from KeyGEN BioTECH (Nanjing, Jiangsu, China). RIPA lysis buffer and the chemiluminescence (ECL) kit were acquired from New Cell & Molecular Biotech Co., Ltd. (Suzhou, Jiangsu, China).

Antibodies against SirT1, p53, HO-1, Caspase-3, Cleaved Caspase-3, Bax, cyclinB1, cyclinE1, CDK2, CDK4, COX4, COX2, NF-κB, p-NF-κB, Cleaved IL-1β, and CX43 were acquired from Cell Signaling Technology, Inc. (Danvers, MA, USA). Antibodies against p21, p16, BCL-2, PGC-1β, and γ-H2AX were acquired from Abcam plc. (Cambridge, England). Antibodies against podoplanin (E11), HSP70, and HSP60 were acquired from Santa Cruz Biotechnology, Inc. (Santa Cruz, CA, USA). Antibodies against Nrf2, FOXO1, SOST, and NLRP3 were acquired from Proteintech Group, Inc. (Wuhan, Hubei, China). Antibodies against COL1A1, osteopontin (OPN), and osteocalcin (OCN) were obtained from ABclonal Technology Co., Ltd. (Wuhan, Hubei, China). Horseradish peroxidase (HRP)-conjugated secondary antibodies were acquired from Jackson ImmunoResearch Inc. (West Grove, PA, USA).

### 2.2. Animals and Experimental Design

Female BALB/c mice (n = 40, 6 weeks old) were acquired from Yangzhou University’s Institute of Comparative Medicine. They were raised in a clean, warm, and comfortable environment (23 ± 1 °C, 12 h light–dark cycle). All the mice were randomly divided into four groups: control (Con), Cd-treated (Cd), OPC-treated (OPC), and combined treatment (Cd + OPC). Animals in the Cd and Cd + OPC groups were given ad libitum drinking water containing CdCl2 (50 mg/L) for 90 days. Mice in the OPC and Cd + OPC groups were given 50 mg/kg OPC administered every two days by oral gavage for 97 days, following a 7-day OPC pretreatment period. After the final treatment and a 12 h fasting period, all animals were euthanized under diethyl ether anesthesia. The tibiae and femora were collected for further studies. All experimental procedures were conducted in accordance with institutional guidelines and approved by the Institutional Animal Care and Use Committee of Yangzhou University (Approval ID: SYXK (Su) 2022-0044).

### 2.3. Histopathological Assessment

The femur samples were decalcified in 10% ethylenediaminetetraacetic acid (EDTA) solution for 30 days at room temperature. The samples were then dehydrated through different concentrations of ethanol solution (70%, 80%, 90%, 95%, and 100%) as previously described [22]. The proximal femoral sections were embedded in paraffin and sectioned to 5 μm thickness using a rotary microtome (Leica, Wetzlar, Germany). The H&E staining kit (Wexis Biotechnology Co., Ltd., Guangzhou, China) was used to stain the slices. Histological images were recorded with a standard Leica inverted microscope (Leica, Wetzlar, Germany).

### 2.4. Immunohistochemical

The paraffin sections of femur were incubated in an oven at 52 °C and then dewaxed in dimethylbenzene. Different concentrations of ethanol solution were further hydrated, and 1% hydrogen peroxide (H_2_O_2_) solution was used to reduce the activity of endogenous oxidases. After that, the slices were incubated with primary antibodies against p16 and SOST overnight at 4 °C, followed by incubation with secondary antibodies for 30 min at 37 °C, and detection of antigen–antibody complexes using 3,3′-diaminobenzidine (DAB). Immunohistochemical images were acquired with a Leica microscope (Leica, Wetzlar, Germany).

### 2.5. Cell Culture and Treatment

MLO-Y4 cells were cultured in α-MEM supplemented with 10% FBS and 1% penicillin–streptomycin solution in an incubator supplied with 5% CO_2_ at 37 °C. To investigate the dose-dependent effects, cells were exposed to varying concentrations of Cd (0, 2, 4, and 6 μmol/L) or OPC (0, 0.25, 0.5, 1, 2, 4, 8, 10, and 20 μmol/L) for 24 h. To investigate the temporal progression of Cd-induced senescence, cells were treated with 6 μmol/L Cd for different times (12, 24, and 48 h). For the positive control experiments, cells were treated with 200 μmol/L H_2_O_2_ for 2 h and then cultured normally for 24 h. For the Nrf2 inhibitor experiments, cells were co-treated with 6 μmol/L Cd and 5 μmol/L ML385 for 24 h. For the protective effect analysis, cells were pretreated with 1 μmol/L OPC for 2 h and then exposed to 6 μmol/L Cd for 24 h.

### 2.6. Cell Viability Assay

The procedure described in Section 2.5 was used to seed cells into 96-well plates at a density of 5 × 10^3^ cells per well. The CCK-8 kit was used to assess cell viability following the manufacturer’s protocol. The absorbance was quantified at 570 nm using an EPOCH microplate spectrophotometer (BioTek Instruments, Inc., Winooski, VT, USA). Additionally, real-time cell proliferation was monitored using the xCELLigence real-time cell analysis (RTCA) system (Roche Applied Science, Basel, Switzerland) as previously described [24].

### 2.7. SA-β-Gal Staining

Staining fixative was used to fix cells for 15 min at 25 °C, followed by three washes with PBS. The SA-β-gal staining solution was incubated with the cells at 37 °C for the entire night. A standard Leica inverted microscope (Leica, Wetzlar, Germany) was used to take the images, and ImageJ 1.42q software (National Institutes of Health, Bethesda, MD, USA) was used to quantify the SA-β-gal positive regions.

### 2.8. ROS Assessment

Intracellular ROS levels were measured using 2′,7′-dichlorofluorescein diacetate (DCFH-DA). Cells were incubated with 10 μM DCFH-DA in serum-free medium for 20 min at 37 °C, followed by three washes with serum-free cell culture medium. ROS levels were detected using BD Biosciences flow cytometry (BD Biosciences, San Jose, CA, USA) or visualized using a Leica laser confocal microscope (Leica, Wetzlar, Germany).

### 2.9. Flow Cytometry

Cellular apoptosis was evaluated using an Annexin V-FITC/PI apoptosis detection kit according to the manufacturer’s instructions. Briefly, cells were co-stained with Annexin V-FITC and PI in binding buffer for 10 min at 25 °C in the dark. The JC-1 kit (Beyotime Biotechnology Inc., Shanghai, China) was used to measure the mitochondrial membrane potential as previously described, with quantification of the red/green fluorescence ratio [25]. For cell cycle analysis, cells were fixed with 70% cold ethanol at 4 °C for 12 h and then stained with PI solution at 37 °C in the dark for 30 min. Flow cytometric analyses were performed using BD Biosciences flow cytometry, and data were processed using FlowJo 10.8.1 software (BD, Franklin Lakes, NJ, USA).

### 2.10. Analysis of Antioxidant Enzymes and Inflammatory Factors

The appropriate kits (Nanjing Jiancheng Bioengineering Co., Ltd., Nanjing, Jiangsu, China) were used to determine the activity of the oxidative enzymes’ total antioxidant capacity (T-AOC), catalase (CAT), glutathione (GSH), and superoxide dismutase (SOD), the protocol was followed as previously described [22], and the OD value was measured using an EPOCH microplate spectrophotometer(Agilent, Santa Clara, CA, USA).

The levels of IL-1, IL-6, and IL-1β in the cells were detected by commercial ELISA kits (Shanghai Enzyme-linked Biotechnology Co., Ltd., Shanghai, China). Cells were collected after experimental treatment and lysed by sonication. The biotin-conjugated detection antibody was added, and the microtiter plate was sealed with an adhesive strip and incubated at 37 °C for 60 min. After incubation, the plate was washed five times with plate wash buffer. After the last wash, Substrate Solutions A and B were added to each well and incubated for 15 min at 37 °C in the dark. After the addition of the termination solution, the OD value at 450 nm was measured using an EPOCH microplate spectrophotometer.

### 2.11. Immunofluorescence

Cells were fixed with 4% paraformaldehyde for 15 min and permeabilized with 0.1% Triton X-100 for 15 min at room temperature. Cells were then blocked with 5% Bovine Serum Albumin (BSA) solution for 30 min. Primary antibodies against p16, γ-H2AX, Nrf2, and NF-κB were incubated at 4 °C overnight, followed by incubation with fluorophore-conjugated secondary antibodies for 2 h at room temperature. Nuclei were stained with DAPI for 10 min. A Leica TCS SP8 STED high-resolution laser confocal microscope (Wetzlar, Germany) was used to capture the images, which were quantitatively analyzed using ImageJ software.

To assess DNA synthesis, cells were labeled using the kFlour488 Click-iT EdU Kit. Cells were incubated with the 10 μmol/L EdU solution for 30 min and processed as previously described [15]. A Leica TCS SP8 STED high-resolution laser confocal microscope (Wetzlar, Germany) was used to capture the images, which were quantitatively analyzed using ImageJ software.

### 2.12. Scanning Electron Microscope (SEM)

For scanning electron microscopy, cells were first fixed with 2.5% glutaraldehyde for 2 h at 4 °C, then post-fixed with 1% osmic acid for 1 h at room temperature. Samples were dehydrated through a graded ethanol series (30%, 50%, 70%, 90%, and 100%), subjected to critical point drying, and sputter-coated with gold-palladium. Ultra-structural images were acquired using a GeminiSEM 300 scanning electron microscope (Carl Zeiss Microscopy, Oberkochen, Germany).

### 2.13. qRT-PCR

Total RNA was extracted with TRIzol according to the manufacturer’s manual. A total of 0.9 µg RNA was used to synthesize cDNA as previously described [22]. qRT-PCR analysis was performed using specific primers (listed in Appendix A) to assess the expression of the following antioxidant response genes: *NRF2*, *NQO1*, *GCLC,* and *GCLM*. Relative gene expression was calculated using the 2^−ΔΔCt^ method, with β-actin serving as the internal reference gene.

### 2.14. Western Blotting

Bone and cellular samples were homogenized in RIPA lysis buffer, and protein concentrations were subsequently normalized using a BCA protein assay kit (Yeasen Biotechnology Co., Ltd., Shanghai, China). Western blot analysis was performed according to previously established protocols [22]. The membranes were incubated with primary antibodies overnight at 4 °C, followed by a 2 h incubation at room temperature with secondary antibodies. Protein bands were visualized using an ECL detection system, and densitometric analysis was conducted using ImageJ software.

### 2.15. Statistical Analysis

Statistical analyses were performed using one-way analysis of variance (ANOVA) with SPSS software (version 26.0; IBM SPSS Inc., Chicago, IL, USA). Differences were considered statistically significant at *p* < 0.05 and extremely significant at *p* < 0.01. All data are presented as the mean ± standard deviation (SD) from at least three independent experiments.

## 3. Results

### 3.1. OPC Alleviates Cd-Induced Cytotoxicity in MLO-Y4 Cells

Cell viability serves as a direct indicator of cytotoxicity. CCK-8 assay and RTCA were performed to investigate the cytotoxicity of Cd in MLO-Y4 cells. Exposure to increasing concentrations of Cd (0, 2, 4, 6, 8, and 10 μmol/L) resulted in a dose-dependent reduction in cell viability (Figure 1B). Correspondingly, the Normalized Cellular Index (NCI) decreased following treatment with different levels of Cd (2, 4, and 6 μmol/L) (Figure 1C). Treatment with OPC (0, 0.25, 0.5, 1, 2, and 4 μmol/L) had no significant effect on cell viability (Figure 1D). However, cell viability was reduced in the higher OPC concentration groups (8, 10, and 20 μmol/L). This dose-dependent response suggests that the effect of OPC may be significantly altered depending on the concentration. Our subsequent experiments showed that 1 μmol/L OPC exhibited the best protective effect (Figure 1E,F). Based on this finding, we drew important conclusions about the optimal concentration range for OPC treatment, and we recommend that lower concentrations (<4 μmol/L) of OPC be used for therapeutic purposes. These quantitative findings were further corroborated by morphological observations (Figure 1G). Collectively, these results demonstrate that OPC exhibits protective effects against Cd-induced cytotoxicity in MLO-Y4 cells in vitro.

### 3.2. OPC Alleviates Cd-Induced Senescence of MLO-Y4 Cells

To explore whether Cd induces senescence of osteocytes in vitro, we used hydrogen peroxide-treated cells as a positive control. We first evaluated cellular senescence markers and found that Cd exposure induced osteocyte senescence (Appendix A). The results of time-course experiments showed that Cd exposure for 24 h induced osteocyte senescence (Appendix A). Next, we evaluated the protective effect of OPC on Cd-induced osteocyte senescence. Compared with the Cd group, SA-β-gal-positive cells were significantly reduced (Figure 2A), the expression of senescence markers (p53, p21, and p16) was reduced, and the expression of the senescence defense protein SirT1 was increased in the Cd + OPC group (Figure 2B). Immunofluorescence analysis showed that the fluorescence intensities of p21 and p16 were significantly reduced after OPC treatment (Figure 2C,D). Collectively, these results demonstrate that OPC can alleviate Cd-induced senescence of MLO-Y4 cells.

### 3.3. OPC Protects MLO-Y4 Cells from Cd-Induced Cell Cycle Arrest and Apoptosis

To investigate the effects of Cd on cell cycle progression and apoptosis in MLO-Y4 cells, flow cytometric analysis was performed. The results revealed that Cd exposure dramatically increased the proportion of cells in the G1 phase while enhancing apoptotic rates (Figure 3A,D). Immunoblotting analysis showed that Cd treatment dramatically elevated the levels of cleaved caspase-3 and the Bax/Bcl-2 ratio, while concurrently reducing the S and G2 phase populations and downregulating the expression of cyclinB1, cyclinE1, CDK2, and CDK4 (Figure 3B,C). Notably, OPC treatment effectively reversed these Cd-induced alterations. These findings suggest that OPC can attenuate Cd-induced G1 phase arrest and the apoptosis of MLO-Y4 cells in vitro.

### 3.4. OPC Alleviates Cd-Induced Mitochondrial Dysfunction and DNA Damage in MLO-Y4 Cells

Mitochondria serve as crucial energy centers that play an essential role in eukaryotic cells and are strongly linked to cell senescence through mitochondrial dysfunction [26]. To investigate the impact of Cd exposure on mitochondrial function, we assessed mitochondrial membrane potential using JC-1 probe analysis. The results revealed that Cd treatment significantly decreased mitochondrial membrane potential, accompanied by the reduced expression of mitochondrial function markers PGC-1β and COX4, and diminished ATP content, while concurrently elevating stress-response proteins HSP70 and HSP60 (Figure 4A–C). Next, we investigated the impact of Cd exposure on DNA replication capacity. The results revealed that Cd treatment significantly reduced the proportion of EdU-positive cells while enhancing both the expression and nuclear distribution of the DNA damage marker γ-H2AX (Figure 4D–F). Notably, OPC treatment effectively ameliorated these Cd-induced alterations. These findings demonstrate that OPC can mitigate Cd-induced mitochondrial dysfunction and DNA damage in MLO-Y4 cells in vitro.

### 3.5. OPC Alleviates Cd-Induced Osteocyte Senescence by Attenuating Oxidative Damage In Vitro

The accumulation of ROS leads to oxidative stress, which is one of the major factors that causes aging [27]. To investigate the protective effects of OPC against Cd-induced oxidative damage in MLO-Y4 cells, we first evaluated intracellular ROS levels. The results revealed that Cd treatment significantly increased the levels of intracellular ROS (Figure 5A,B). Cellular oxidative stress is controlled by the transcription factor Nrf2 [28]. Cd treatment significantly upregulated the expression of Nrf2, HO-1, and the mRNA levels of *Nrf2*, *NQO1*, *GCLC*, and *GCLM* (Figure 5C,D). The immunofluorescence results showed that Cd treatment significantly increased the nuclear translocation of Nrf2 (Figure 5E,G). Furthermore, Cd significantly decreased the activities of CAT, GSH, SOD, and the capacity of T-AOC (Figure 5F). Notably, OPC pretreatment effectively attenuated these Cd-induced alterations in oxidative stress parameters. These findings suggest that OPC can mitigate Cd-induced ROS accumulation and oxidative damage in vitro.

To further explore the role of the Nrf2 signaling pathway in Cd-induced osteocyte senescence, we used Nrf2 inhibitor ML385 to inhibit Nrf2 expression and assess cell viability. The results showed that 5 μmol/L ML385 treatment significantly inhibited Nrf2 expression and had no significant effect on cell viability (Appendix A). As shown in Appendix A, ML385 treatment reversed the Cd-increased protein expression of Nrf2 and HO-1. Next, we found that ML385 significantly reduced Cd-induced osteocyte senescence (Figure 5H–M). The results suggest that the activation of the Nrf2 signaling pathway plays a negative role in Cd-induced osteocyte senescence.

### 3.6. In Vitro, OPC Inhibits Cd-Induced SASP Synthesis

Chronic inflammation represents a significant endogenous driver of cellular aging, and its modulation presents a potential therapeutic strategy for age-related conditions [29]. The transcription factors of the NF-κB family are important modulators of inflammation and aging processes [30]. Analysis of inflammatory markers revealed that Cd exposure markedly increased the levels of IL-1, IL-1β, and IL-6 in MLO-Y4 cells, concurrent with enhanced expression of the inflammation-related proteins NLRP3, Cleaved caspase-1, COX2, Cleaved IL-1β, and the phosphorylation of NF-κB (Figure 6A–C). Immunofluorescence analysis further demonstrated that Cd treatment enhanced the nuclear translocation of NF-κB (Figure 6D). Notably, OPC treatment effectively suppressed these Cd-induced inflammatory responses. These findings demonstrate that OPC can suppress the Cd-triggered inflammatory response by inhibiting NF-κB signaling in vitro.

### 3.7. In Vitro, OPC Against Cd-Induced Damage of Dendritic Synapses in MLO-Y4 Cells

Given that osteocytes maintain bone homeostasis through their extensive dendritic network, we investigated the impact of Cd exposure on osteocyte connectivity and dendrite-associated protein expression. SEM analysis revealed that Cd treatment significantly disrupted dendritic network integrity (Figure 7A). Western blot analysis revealed that Cd treatment significantly disrupted dendritic network integrity and downregulated key proteins involved in osteocyte communication and function, including the dendrite marker E11, gap junction protein CX43, and bone matrix proteins (COL1A1, OPN, OCN, and SOST) in MLO-Y4 cells (Figure 7B,C). Notably, OPC pretreatment effectively preserved dendritic network architecture and maintained the expression levels of these crucial osteocyte-specific proteins. Collectively, these findings demonstrate that OPC exhibits protective effects against Cd-induced disruption of osteocyte dendritic networks and associated protein expression.

### 3.8. OPC Alleviates Cd-Induced Osteocyte Senescence In Vivo

To evaluate the protective effects of OPC against Cd-induced osteocyte senescence in vivo, we established a mouse model with chronic Cd exposure and OPC supplementation over a 90-day period (Figure 8A). Histological analysis revealed that Cd exposure led to an increase in the number of blank osteocyte compartments in the femur, indicating a reduction in the total number of osteocytes, while OPC supplementation relieved this result (Figure 8B). Immunohistochemical and Western blot analysis demonstrated that OPC treatment preserved the population of SOST-positive osteocytes and maintained the expression of key osteocyte-function proteins, including COL1A1, OPN, OCN, SOST, and E11 (Figure 8C,D). Furthermore, OPC significantly reduced both the accumulation of the senescence marker p16 in femoral osteocytes and the expression levels of senescence-associated proteins p53, p21, and p16 (Figure 8E,F). These results demonstrate that OPC effectively attenuates Cd-induced osteocyte senescence in vivo.

## 4. Discussion

Cd, as a heavy metal contaminant, exhibits bioaccumulation through the food chain and possesses a prolonged biological half-life, with toxicity even at low-dose exposures [1]. The skeletal system represents a primary target of Cd toxicity, leading to pathological conditions including osteoporosis and osteomalacia [31,32]. Several studies have elucidated the fundamental impacts of Cd on osteoclasts and osteoblasts [5,7]. However, the osteotoxicity of Cd on osteocytes has not been fully confirmed. Moreover, there is a paucity of effective prophylactic agents targeting Cd-induced osteocyte senescence, particularly natural compounds. OPC is a naturally occurring polyphenolic flavonoid compound that demonstrates comprehensive pharmacological activities [33,34]. In the present investigation, a Cd-induced osteocyte senescence model was performed in vitro and in vivo. We found that OPC effectively attenuates this process by inhibiting oxidative damage and inflammatory responses. Mechanistic analyses revealed that OPC modulates oxidative stress and inflammation via the inhibition of the Nrf2 and NF-κB signaling pathways, thereby establishing a novel therapeutic mechanism for OPC in ameliorating Cd-induced osteotoxicity.

Cellular senescence can be induced by multiple factors, encompassing both intrinsic mediators, such as oxidative stress and mitochondrial dysfunction, and extrinsic stimuli, including ionizing radiation and chemotherapeutic agents. These diverse triggers converge on the activation of the p53/p21/p16 signaling cascade through DNA damage induction, culminating in cell cycle arrest [35]. The cell cycle progression is a strict regulatory mechanism that involves the division, proliferation, and differentiation of cells, primarily regulated by the retinoblastoma tumor suppressor protein (RB) and cyclin-dependent kinases (CDKs) [36]. Cyclin–CDK activity is notably suppressed in senescent cells, as CDK inhibitors p21, p15, and p16 have a significant part in senescence signaling [37]. Previous studies have demonstrated that Cd exposure induces senescence in bone marrow stromal cells (BMSCs) through the activation of DNA damage responses, cell cycle arrest, and enhanced expression of p53, p21, and p16 [15]. Our current findings reveal that Cd exposure triggers G0/G1 phase arrest, enhances β-gal activity, and upregulates p53, p21, and p16 expression in osteocytes. Furthermore, DNA damage is an important trigger of senescence, whereby ineffective DNA repair or mismatch repair mechanisms lead to cell cycle arrest, ultimately triggering senescence [38,39]. Recent studies have demonstrated that osteoprogenitor cells from elderly mice showed signs of senescence and DNA damage [40]. Our experimental results indicate that OPC co-treatment substantially attenuates Cd-induced cell cycle arrest and DNA damage. Prior research has indicated that OPC is one of the most efficacious cytoprotective agents against various adverse condition-induced DNA damage [41,42,43]. The present study provides the first evidence demonstrating OPC’s protective effects against Cd-induced DNA damage in osteocytes.

Mitochondria are the fundamental organelles regulating energy in eukaryotic cells and are the primary source of intracellular ROS [44]. Senescent cells show signs of elevated ROS levels and decreased mitochondrial membrane potential [45]. Furthermore, chronic DNA damage is caused by mitochondrial malfunction in senescent cells via the nucleus–mitochondria and mitochondria–nucleus pathways [46]. In a recent investigation, Sirt4-deficient chondrocytes exhibited enhanced senescent phenotypes, characterized by decreased ATP generation and a lower mitochondrial membrane potential, whereas Sirt4 overexpression in an osteoarthritis (OA) mouse model effectively restored mitochondrial function and prevented chondrocyte senescence [47]. Our findings demonstrate that OPC confers protective effects against Cd-induced mitochondrial dysfunction in osteocytes, specifically preserving membrane potential. Moreover, the accumulation of oxidatively modified proteins within mitochondria contributes substantially to functional decline, a process critically regulated by molecular chaperones including HSP60 and HSP70 [48]. Previous studies in avian cerebral tissue have established that Cd exposure significantly upregulates both mRNA and the protein expression of heat shock proteins [49]. Our results demonstrated that OPC effectively suppresses the Cd-induced elevation of HSP60 and HSP70 expression. This suggests that OPC can prevent oxidized proteins from aggregating in the mitochondria, and the mechanism for this effect may be connected to OPC’s ability to remove ROS.

Oxidative stress represents a major pathway for Cd-induced cytotoxicity. In testicular tissue, Cd exposure causes a lower GSH concentration and increased generation of ROS, thereby compromising spermatogenesis and testicular development [50]. In bone tissue, Cd exposure induces osteoblast apoptosis through oxidative stress and ultimately leads to osteoporosis [51]. Our experimental results showed that OPC co-treatment significantly alleviated Cd exposure-induced ROS accumulation in osteocytes and restored antioxidant enzyme activities. The cellular response to oxidative stress is primarily mediated through Nrf2, a transcription factor that translocates to the nucleus, activating downstream transcription factors [52]. Previous research has demonstrated that the modulation of Nrf2-mediated oxidative stress is an effective strategy to mitigate Cd-induced oxidative damage in both neuronal tissue [53] and hepatic tissue [54]. Concordantly, our study showed that OPC significantly reversed the Cd-induced activation of the Nrf2 pathway. Furthermore, we found that the pharmacological inhibition of the Nrf2 pathway reverses Cd-induced osteocyte senescence. This finding demonstrates that the overactivation of the Nrf2 pathway plays a negative role in osteocyte senescence, providing strong evidence for a causal relationship between the protective effect of OPC and the Nrf2 pathway.

The aging process is characterized by systemic chronic inflammation, with emerging evidence establishing inflammation as an intrinsic mediator of age-related physiological decline. Consequently, therapeutic strategies targeting inflammatory pathways have emerged as promising interventions in age-related pathologies [29]. Previous research has demonstrated that Cd exposure initiates the activation of the NF-κB signaling pathway and NLRP3 inflammatory vesicles in renal tubular epithelial cells, leading to enhanced secretion of pro-inflammatory cytokines, such as TNF-α and IL-1β, ultimately precipitating inflammatory injury in the rat kidney [55]. Furthermore, compelling evidence indicates that OPC-rich substance prolonged the lifespan of the senescence-accelerated mouse model (SAMP8) by modulating the severity of infection-dependent inflammation [56]. In our work, Cd exposure significantly elevated the production of IL-1, IL-6, and IL-1β in osteocytes, concurrent with the activation of the NF-κB pathway. Notably, OPC significantly reversed the Cd-induced inflammatory response. Collectively, these findings demonstrate that OPC exhibits protective effects against Cd-induced inflammatory responses in osteocytes through the targeted inhibition of the NF-κB pathway.

Osteocytes constitute the predominant proportion of adult bone cells, although historically these cells were considered passive and metabolically quiescent entities with minimal contribution to bone growth and development [4]. Contemporary research has fundamentally established osteocytes as critical regulators of skeletal homeostasis [57]. Indeed, osteocytes are involved in bone aging, and a characteristic of aged skeletons is the progressive decline in osteocyte density [58]. Osteocytes have a highly dendritic morphology, and a large number of dendritic protrusions connect osteocytes into a functional network system. Another striking characteristic of aged osteocytes is the significant reduction in dendritic processes, substantially compromising intercellular communication within the osteocyte network [59]. Our results demonstrated that exposure to Cd significantly diminishes osteocyte populations both in vivo and in vitro. Furthermore, we found that Cd exposure induces dendritic retraction and a loss of intercellular synapses between MLO-Y4 cells, potentially compromising osteocyte signal conduction and skeletal homeostasis. Collectively, these findings demonstrate that the OPC intervention represents an efficacious therapeutic strategy for mitigating cadmium-induced reductions in bone cell density and associated functional impairments.

## 5. Conclusions

In conclusion, our study demonstrates that OPC effectively attenuates Cd-induced premature senescence in osteocytes by suppressing oxidative stress, inflammatory response, and DNA damage (Figure 9). There are still some limitations in our research, and it is necessary to use genetic approaches to further validate the relevant mechanisms in the future. Overall, this study provides the first evidence of Cd-induced osteocyte senescence and dysfunction, and establishes OPC as a promising therapeutic agent for preventing Cd-mediated osteotoxicity. These findings not only improve our understanding of the molecular mechanisms of cadmium-induced bone disease, but also provide a foundation for the development of interventions based on natural compounds.

## Figures and Tables

**Figure 1 antioxidants-13-01515-f001:**
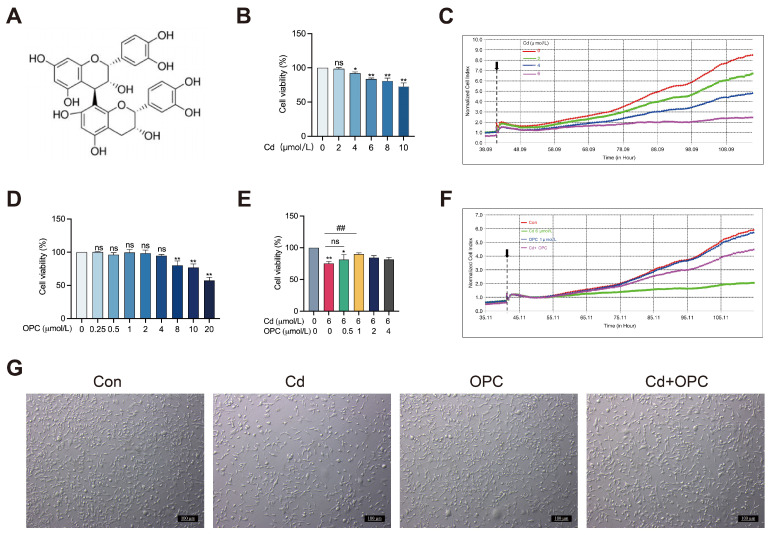
OPC alleviates Cd-induced cytotoxicity in MLO-Y4 cells. The chemical structure of OPC (**A**), MLO-Y4 cells were exposed to various concentrations of Cd or OPC for 24 h. Cell viability was detected using CCK8 assay (**B**,**D**) and RTCA (**C**) (The black arrow indicates the point in time when Cd exposure began). MLO-Y4 cells were pretreated with OPC for 2 h, followed by exposure to 6 μmol/L Cd for 24 h. Cell viability was evaluated using CCK-8 assay (**E**) and RTCA (**F**) (The black arrow indicates the point at which the Cd treatment began), and morphological changes were examined by phase-contrast microscopy (**G**). Scale bar = 100 μm. Results are shown as mean ± SD (n = 3). Compared with the control group, * *p* < 0.05, ** *p* < 0.01, or ns represent *p* > 0.05. Compared with the Cd group, ^##^
*p* < 0.01, or ns represent *p* > 0.05.

**Figure 2 antioxidants-13-01515-f002:**
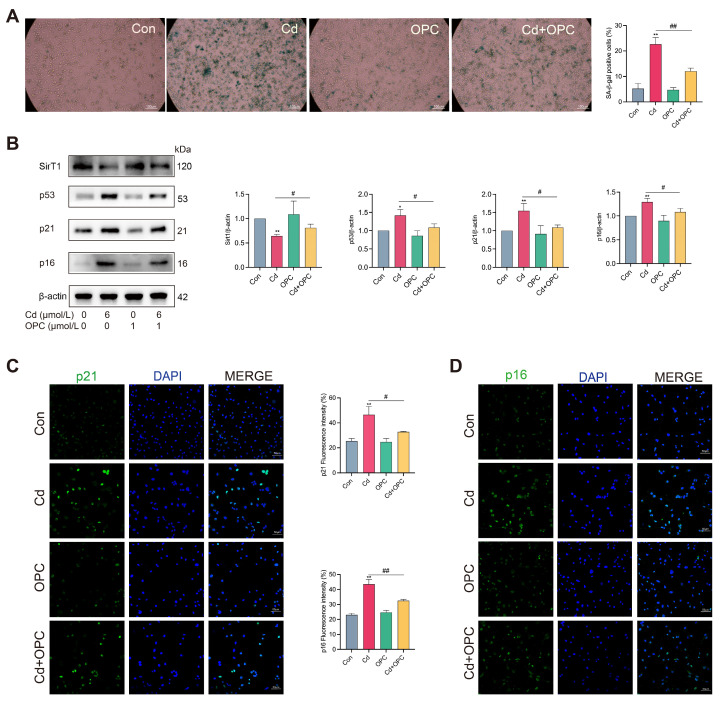
OPC alleviates Cd exposure-induced cellular senescence. MLO-Y4 cells were pretreated with 1 μmol/L OPC for 2 h, followed by exposure to 6 μmol/L Cd for 24 h. Cellular senescence was detected by SA-β-gal staining (**A**). Scale bar = 100 μm. The protein expression of SirT1, p53, p21, and p16 was detected by Western blot (**B**). p21 and p16 fluorescence intensities in MLO-Y4 cells were observed using immunofluorescence and quantitatively analyzed using ImageJ software (**C**,**D**). Scale bar = 50 μm. Results are shown as the mean ± SD (n = 3). Compared with the control group, * *p* < 0.05, ** *p* < 0.01. Compared with the Cd group, ^#^
*p* < 0.05, ^##^
*p* < 0.01.

**Figure 3 antioxidants-13-01515-f003:**
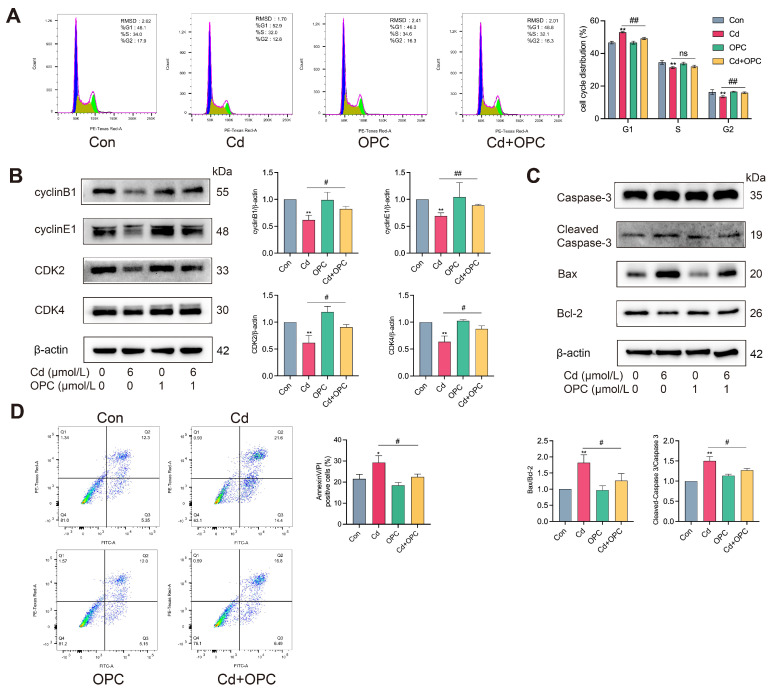
OPC alleviated Cd-induced cell cycle arrest and apoptosis**.** MLO-Y4 cells were pretreated with 1 μmol/L OPC for 2 h, followed by exposure to 6 μmol/L Cd for 24 h. Cell cycle distribution was detected by flow cytometry (**A**). The protein expression of cyclinB1, cyclinE1, CDK2, CDK4, caspase-3, Bax, BCL-2 and the levels of cleaved caspase-3 were detected by Western blot (**B**,**C**). Cell apoptosis was detected by flow cytometry (**D**). Results are shown as the mean ± SD (n = 3). Compared with the control group, * *p* < 0.05, ** *p* < 0.01, or ns represent *p* > 0.05. Compared with the Cd group, ^#^
*p* < 0.05, ^##^
*p* < 0.01, or ns represent *p* > 0.05.

**Figure 4 antioxidants-13-01515-f004:**
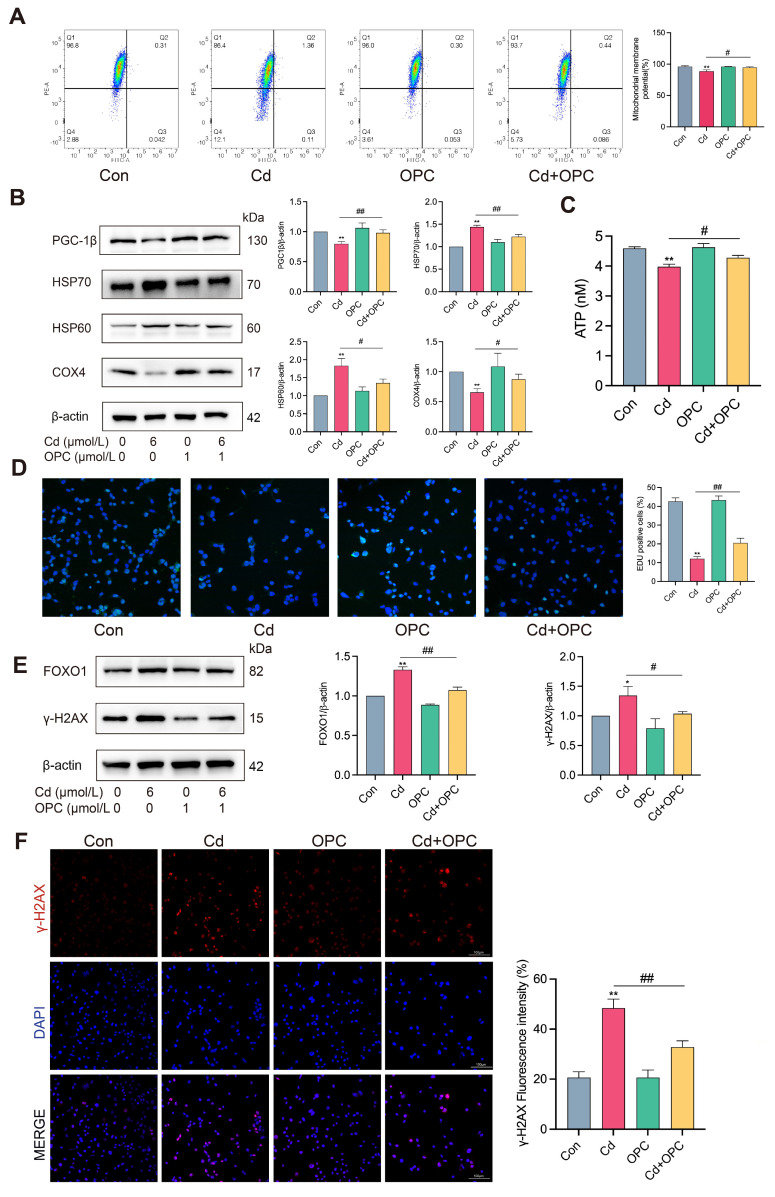
OPC alleviates Cd-induced mitochondrial dysfunction and DNA damage. MLO-Y4 cells were pretreated with 1 μmol/L OPC for 2 h, followed by exposure to 6 μmol/L Cd for 24 h. Mitochondrial membrane potential was detected by the JC-1 probe using flow cytometry (**A**). The protein expression of PGC-1β, HSP70, HSP60, and COX4 was detected by Western blot (**B**). ATP content were detected by ATP assay kit (**C**). DNA replication ability was detected by EdU staining (**D**). Scale bar = 50 μm. The expression of FOXO1 and γ-H2AX was detected by Western blot (**E**). Immunofluorescence analysis showed nuclear distribution and intensity of γ-H2AX foci (**F**). Scale bar = 100 μm. Results are shown as the mean ± SD (n = 3). Compared with the control group, * *p* < 0.05, ** *p* < 0.01. Compared with the Cd group, ^#^
*p* < 0.05, ^##^
*p* < 0.01.

**Figure 5 antioxidants-13-01515-f005:**
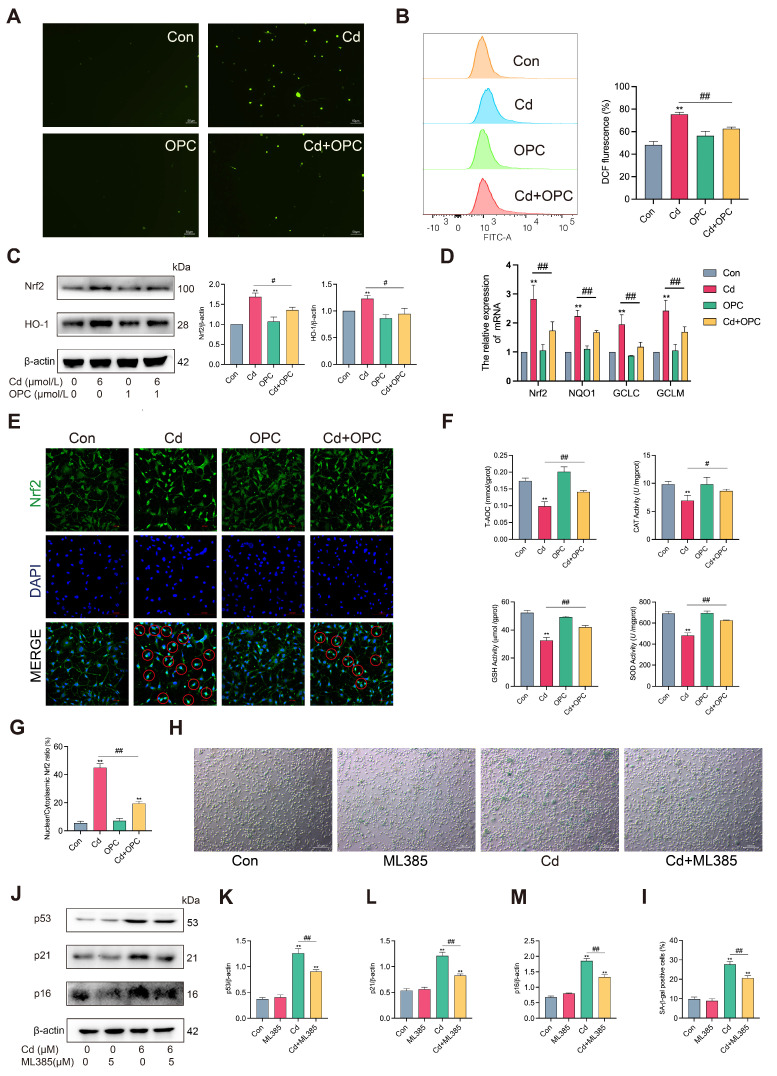
OPC alleviates Cd-induced ROS accumulation and oxidative damage by inhibiting the Nrf2 pathway. MLO-Y4 cells were pretreated with 1 μmol/L OPC for 2 h, followed by exposure to 6 μmol/L Cd for 24 h. ROS levels were observed using fluorescence microscopy (**A**) or detected using flow cytometry (**B**). The expression of Nrf2 and HO-1 was detected by Western blot (**C**). The mRNA levels of *Nrf2*, *HO-1*, *GCLC*, and *GCLM* were detected using qRT-PCR (**D**). Nrf2 nuclear translocation in MLO-Y4 cells was observed using immunofluorescence, red circles represent cells with Nrf2 nuclear translocation; specifically, set a fluorescence intensity threshold value, above which cells will be considered to have positive nuclear translocation of Nrf2, then calculate the percentage of cells with positive nuclear translocation for each treatment group based on the established threshold (**E**,**G**). (Red circles represent cells with positive Nrf2 nuclear translocation). Scale bar = 100 μm. The activities of T-AOC, CAT, GSH, and SOD were determined using colorimetry (**F**). MLO-Y4 cells were co-treated with 6 μmol/L Cd and 5 μmol/L ML385 for 24 h. Cellular senescence was detected by SA-β-gal staining (**H**,**I**). Scale bar = 100 μm. The protein expression of p53, p21, and p16 was detected by Western blot (**J**–**M**). Results are shown as the mean ± SD (n = 3). Compared with the control group, ** *p* < 0.01. Compared with the Cd group, ^#^
*p* < 0.05, ^##^
*p* < 0.01.

**Figure 6 antioxidants-13-01515-f006:**
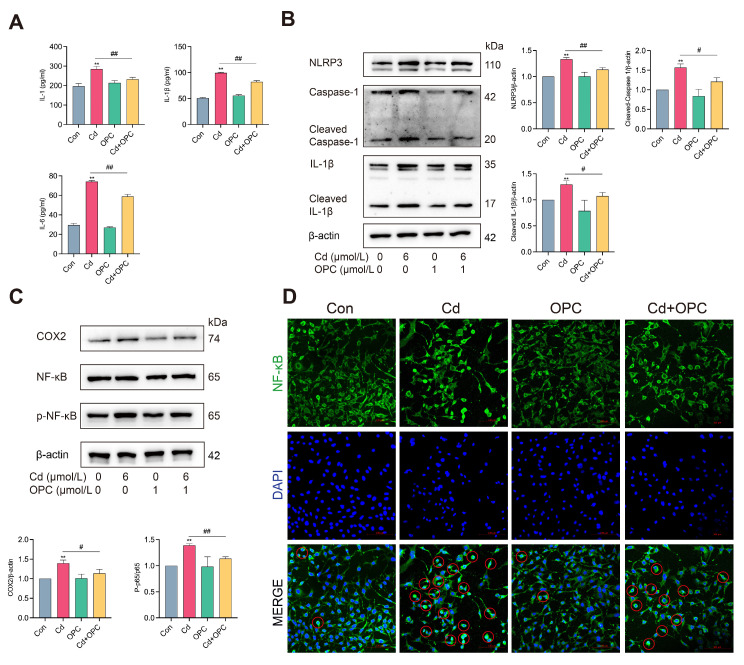
OPC reduces Cd-induced SASP production by inhibiting the NF-κB pathway. MLO-Y4 cells were pretreated with 1 μmol/L OPC for 2 h, followed by exposure to 6 μmol/L Cd for 24 h. The levels of IL-1, IL-1β, and IL-6 intracellularly were detected by ELISA kits (**A**). The expression of NLRP3, Cleaved Caspase-1, Cleaved IL-1β, COX2, NF-κB, and p-NF-κB was detected by Western blot (**B**,**C**). NF-κB nuclear translocation in MLO-Y4 cells was observed using immunofluorescence, red circles represent cells with NF-κB nuclear translocation (**D**). (Red circles represent cells with positive NF-κB nuclear translocation). Scale bar = 100 μm. Results are shown as the mean ± SD (n = 3). Compared with the control group, ** *p* < 0.01. Compared with the Cd group, ^#^
*p* < 0.05, ^##^
*p* < 0.01.

**Figure 7 antioxidants-13-01515-f007:**
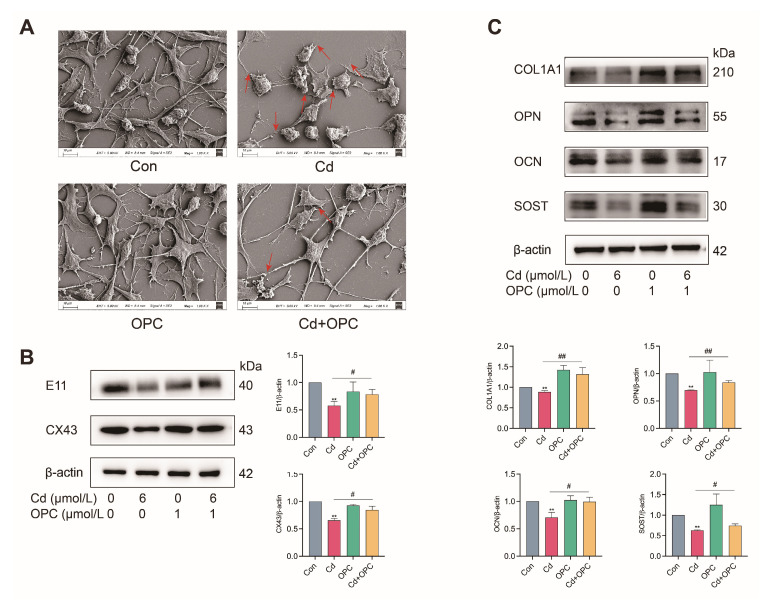
OPC protects Cd-induced damage to dendritic synapses between MLO-Y4 cells. MLO-Y4 cells were pretreated with 1 μmol/L OPC for 2 h, followed by exposure to 6 μmol/L Cd for 24 h. Intercellular synaptic structures were observed by SEM (**A**).(The red arrows represent damaged or broken synapses). Scale bar = 10 μm. The expression of E11, CX43, COL1A1, OPN, OCN, and SOST was detected by Western blot (**B**,**C**). Results are shown as the mean ± SD (n = 3). Compared with the control group, ** *p* < 0.01. Compared with the Cd group, ^#^
*p* < 0.05, ^##^
*p* < 0.01.

**Figure 8 antioxidants-13-01515-f008:**
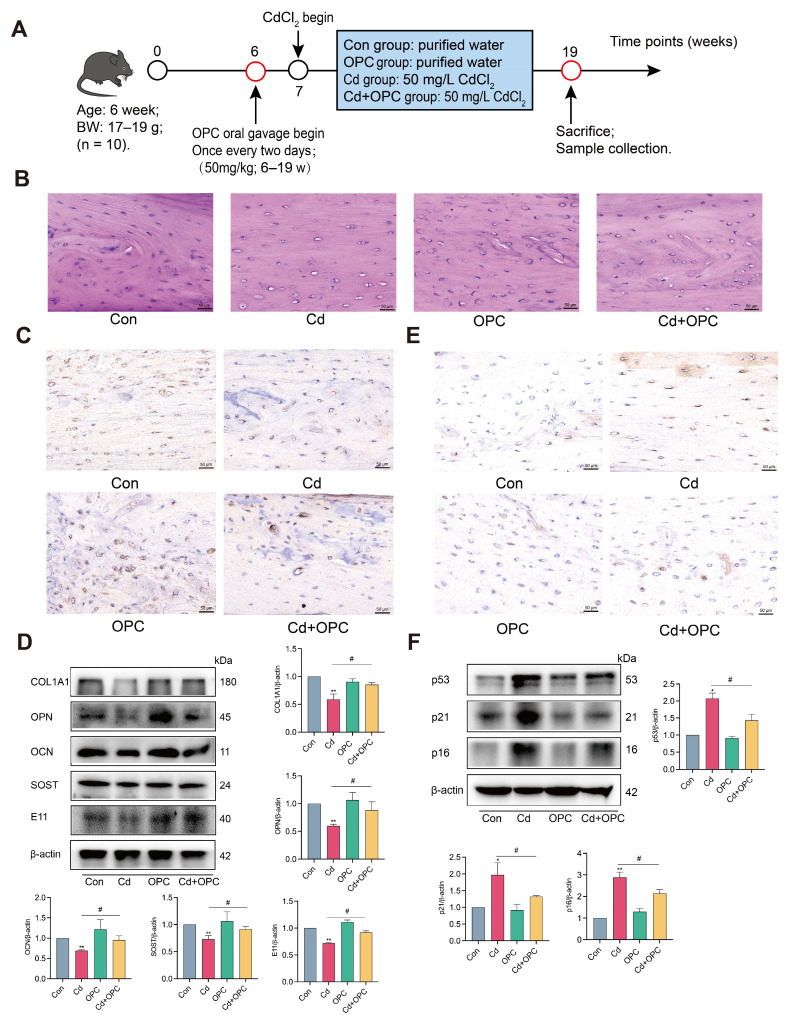
OPC attenuates Cd-induced osteocyte senescence and dysfunction in vivo. Schematic diagram of the animal experiment design and treatment protocol (**A**). Histopathological evaluation of femur was conducted by HE staining (**B**), scale bar = 50 μm. SOST (**C**) and p16 (**E**) expression in femur osteocytes was observed using IHC staining, scale bar = 50 μm. The expression of osteocyte- function proteins COL1A1, OPN, OCN, SOST, E11 (**D**) and senescence-associated proteins p53, p21, p16 (**F**) in the femur was detected by Western blot. Results are shown as the mean ± SD (n = 3). Compared with the control group, * *p* < 0.05, ** *p* < 0.01. Compared with the Cd group, ^#^
*p* < 0.05.

**Figure 9 antioxidants-13-01515-f009:**
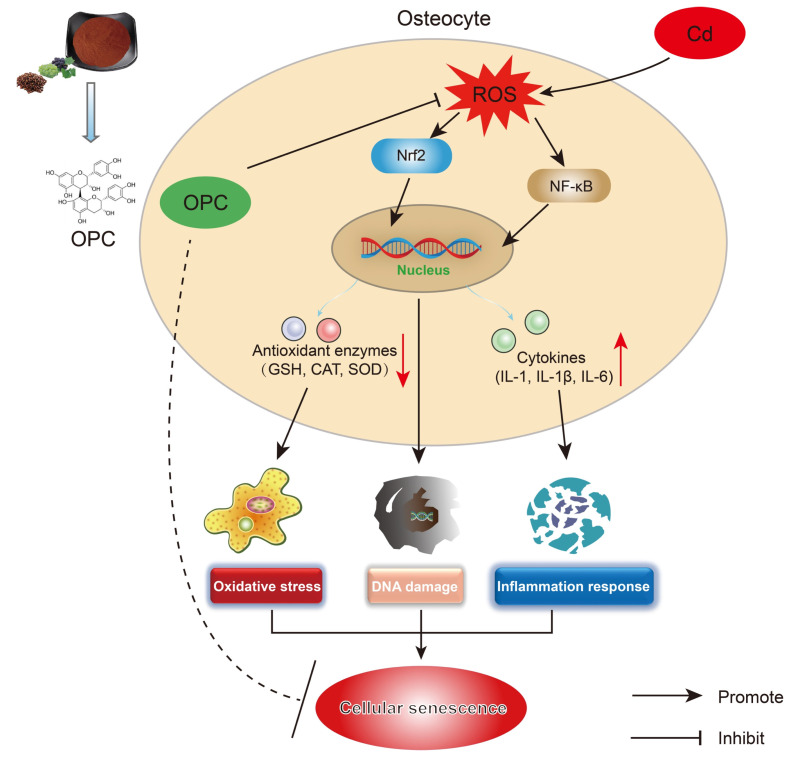
Schematic representation of the dual molecular mechanisms by which OPC ameliorates cadmium-induced senescence of osteocytes: (i) suppression of oxidative stress and ROS accumulation through modulating the Nrf2 signaling pathway, and (ii) inhibition of pro-inflammatory cytokine production via regulating the NF-κB signaling. (Red downward arrows represent decreased levels of antioxidant enzyme activity, and red up arrows represent increased levels of inflammatory factors).

## Data Availability

The data presented in this study are available on request from the corresponding authors.

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
