# Peer review of "Oligomeric Proanthocyanidins Ameliorate Cadmium-Induced Senescence of Osteocytes Through Combating Oxidative Stress and Inflammation"

_antioxidants, 2024, doi:10.3390/antiox13121515_

Round 1
Reviewer 1 Report
1. Section 2.2 – Authors should include a copy of the signed approval for animal testing; I suggest putting it in supplementary materials.
2. Conclusions – I appreciate the figure summarising the research conducted; however, the section should also contain at least a short comment from the authors summarizing the study.
1. Line 62 – SASP abbreviation should be explained
2. Line 73, 447 – in vitro – italic

Author Response
Response to the Reviewer
Dear Reviewer,
Thanks a lot for your kindly reviewing of the manuscript entitled “Oligomeric proanthocyanidins ameliorate cadmium-induced senescence of osteocytes through combating oxidative stress and inflammation” (Manuscript Number antioxidants-3278416). On behalf of the co-authors, I am highly thankful to you for giving us an opportunity to revise our manuscript. We appreciate your constructive comments and suggestions on our manuscript. Those comments are valuable and very helpful for revising and improving our manuscript work quality.
We have tried our best to revise our manuscript after considering your comments carefully and have made revision marked in yellow in the manuscript. We have revised the introduction section, in particular, we have added research background related to senescence. Attached is the point-by–point response to your comments and suggestions. We would like to express our great appreciation to you for comments on our paper. If you have any question about the manuscripts, please let me know.
Thank you and best regards.
Yours sincerely,
Zongping Liu and Xishuai Tong
Here below is our description on revisions according to the reviewers’ comments point by point:
Major comments
- Section 2.2 – Authors should include a copy of the signed approval for animal testing; I suggest putting it in supplementary materials.
Response: Thank you for your advice. We have put the signed approval for animal testing in supplementary materials.
- Conclusions – I appreciate the figure summarising the research conducted; however, the section should also contain at least a short comment from the authors summarizing the study.
Response: Thank you for your advice. We have added a short comment below the graphic abstract.
- Cell Culture Model:The use of MLO-Y4 osteocyte cells is appropriate for studying osteocyte function in vitro. However, the addition of other cell lines or the use of primary osteocytes could provide further validation of the findings in more complex systems
Response: Thank you for your advice. We have tried to extract primary osteocytes, but it is difficult to extract enough osteocytes for subsequent experiments because osteocytes are present in cortical bone. We will verify this result with other cell lines in the next experiment. Thank you very much for your many useful suggestions for our future research.
- Precision in Experimental Conditions:The procedures are generally well described, but some details—such as specific incubation times or the standardization/calibration of instruments—are lacking. Providing these details would enhance the reproducibility of the experiments for other researchers
Response: Thank you for your advice. We have added more detailed experimental information in the Materials and Methods section.
- Time Points: Adding more time points in both in vivo and in vitro experiments would help clarify how processes such as apoptosis, oxidative stress, and inflammation evolve with Cd and OPC exposure. This would also allow for an assessment of whether the effects of OPC are sustained or transient.
Response: Thank you for your advice. We acknowledge that additional time points would provide valuable insights into the dynamic nature of these processes. Our current time points were carefully selected based on previous literature showing peak responses for these parameters. We will add more time points to elucidate the effects of different times on cadmium toxicity and OPC treatment efficacy in the next study. Thank you very much for your many useful suggestions for our future research.
- The references are appropriate and cite relevant studies in Cd toxicity, osteocyte biology, oxidative stress, and inflammation. However, including more recent studies on the role of polyphenols and antioxidants in osteocyte function could further support the novelty of the research.
Response: Thank you for your advice. We add the latest research on the role of polyphenols and antioxidants in osteocyte function.
Detail comments
- Line 62 – SASP abbreviation should be explained
Response: Thank you for your advice. We corrected it.
- Line 73, 447 – in vitro – italic
Response: Thank you for your advice. We corrected it.

Reviewer 2 Report
Review comments for antioxidants-3278416.
In this manuscript, the authors examined the effects of oligomeric proanthocyanidins on Cadmium-induced senescence of osteocyte. Using both in vitro (MLO-Y4 cells) and in vivo (mouse) models, the authors demonstrate that OPC treatment attenuates cadmium-induced oxidative stress and inflammatory responses in osteocytes. They propose that this protection occurs through modulation of the Nrf2 and NF-kappaB signaling pathways. The authors observe that OPC treatment reduces senescence markers, preserves osteocyte dendritic networks, and maintains bone cell function. While presenting an interesting therapeutic possibility for preventing heavy metal-induced bone aging, the study lacks crucial mechanistic validation and physiologically relevant dosing studies that would be necessary to establish OPC as a viable therapeutic agent.
Here are comments.
Major comments:
1. Cadmium dosing
Why the authors used relatively low dose of cadmium (6 microM) in this study? Several studies used higher concentration of cadmium such as 50 microM. The authors should discuss on the difference in the concentration.
2. Mechanistic Validation
The authors claim OPC works through Nrf2 and NF-kappaB pathways, but no mechanistic validation using pathway inhibitors or genetic approaches (e.g., siRNA knockdown) was performed.
- Direct evidence showing OPC-Nrf2 or OPC-NF-kappaB interaction is lacking.
- Additional experiments using Nrf2 or NF-kappaB knockout models would strengthen the proposed mechanism.
3. Technical Issues
- Critical controls are missing, such as positive controls for senescence markers.
- The study lacks proper aging controls to distinguish between natural and Cd-induced senescence.
- No quantitative analysis of osteocyte dendritic network changes was provided.
4. Data Presentation
- Quantification of immunofluorescence data is missing.
5. Experimental Design
- Western blotting for Nrf2 should use nuclear faction as sample to clarify the extent of nuclear translocation.
- The 90-day treatment period seems arbitrary and requires justification.
- No time-course experiments were performed to track the progression of senescence.
- The study lacks investigation of potential side effects of long-term OPC treatment.
6. Translational Potential
- The authors do not address the bioavailability of OPC.
- No pharmacokinetic data is presented.
- The potential clinical applications need more thorough discussion.
Minor Comments:
1. Several figure legends lack sufficient detail.
2. Methods section requires more detailed protocols.
3. Some statistical analyses need clarification.
Review comments for antioxidants-3278416.
In this manuscript, the authors examined the effects of oligomeric proanthocyanidins on Cadmium-induced senescence of osteocyte. Using both in vitro (MLO-Y4 cells) and in vivo (mouse) models, the authors demonstrate that OPC treatment attenuates cadmium-induced oxidative stress and inflammatory responses in osteocytes. They propose that this protection occurs through modulation of the Nrf2 and NF-kappaB signaling pathways. The authors observe that OPC treatment reduces senescence markers, preserves osteocyte dendritic networks, and maintains bone cell function. While presenting an interesting therapeutic possibility for preventing heavy metal-induced bone aging, the study lacks crucial mechanistic validation and physiologically relevant dosing studies that would be necessary to establish OPC as a viable therapeutic agent.
Here are comments.
Major comments:
1. Cadmium dosing
Why the authors used relatively low dose of cadmium (6 microM) in this study? Several studies used higher concentration of cadmium such as 50 microM. The authors should discuss on the difference in the concentration.
2. Mechanistic Validation
The authors claim OPC works through Nrf2 and NF-kappaB pathways, but no mechanistic validation using pathway inhibitors or genetic approaches (e.g., siRNA knockdown) was performed.
- Direct evidence showing OPC-Nrf2 or OPC-NF-kappaB interaction is lacking.
- Additional experiments using Nrf2 or NF-kappaB knockout models would strengthen the proposed mechanism.
3. Technical Issues
- Critical controls are missing, such as positive controls for senescence markers.
- The study lacks proper aging controls to distinguish between natural and Cd-induced senescence.
- No quantitative analysis of osteocyte dendritic network changes was provided.
4. Data Presentation
- Quantification of immunofluorescence data is missing.
5. Experimental Design
- Western blotting for Nrf2 should use nuclear faction as sample to clarify the extent of nuclear translocation.
- The 90-day treatment period seems arbitrary and requires justification.
- No time-course experiments were performed to track the progression of senescence.
- The study lacks investigation of potential side effects of long-term OPC treatment.
6. Translational Potential
- The authors do not address the bioavailability of OPC.
- No pharmacokinetic data is presented.
- The potential clinical applications need more thorough discussion.
Minor Comments:
1. Several figure legends lack sufficient detail.
2. Methods section requires more detailed protocols.
3. Some statistical analyses need clarification.
Author Response
Response to the Reviewer
Dear Reviewer,
Thanks a lot for your kindly reviewing of the manuscript entitled “Oligomeric proanthocyanidins ameliorate cadmium-induced senescence of osteocytes through combating oxidative stress and inflammation” (Manuscript Number antioxidants-3278416). On behalf of the co-authors, I am highly thankful to you for giving us an opportunity to revise our manuscript. We appreciate your constructive comments and suggestions on our manuscript. Those comments are valuable and very helpful for revising and improving our manuscript work quality.
We have tried our best to revise our manuscript after considering your comments carefully and have made revision marked in yellow in the manuscript. We have re-edited the article, especially in the results and figures section. Attached is the point-by–point response to your comments and suggestions. We would like to express our great appreciation to you for comments on our paper. If you have any question about the manuscripts, please let me know.
Thank you and best regards.
Yours sincerely,
Zongping Liu and Xishuai Tong
Here below is our description on revisions according to the reviewers’ comments point by point:
Major comments:
- Cadmium dosing
Why the authors used relatively low dose of cadmium (6 microM) in this study? Several studies used higher concentration of cadmium such as 50 microM. The authors should discuss on the difference in the concentration.
Response: Thank you for this important question about cadmium concentration. We selected 6 μM Cd on several key considerations: This concentration better reflects actual blood cadmium levels in environmentally exposed populations, whereas 50 μM represents acute toxicity scenarios. Our preliminary experiments showed that 6 μM Cd induced significant cellular responses while maintaining cell viability at around 70%. Higher concentrations caused excessive cell death, which would have hindered our investigation of OPC's protective mechanisms. Our study focused on investigating protective mechanisms under moderate stress conditions that mimic environmental exposure rather than acute toxicity. We hope that our corrections will satisfy you.
- Mechanistic Validation
The authors claim OPC works through Nrf2 and NF-kappaB pathways, but no mechanistic validation using pathway inhibitors or genetic approaches (e.g., siRNA knockdown) was performed.
- Direct evidence showing OPC-Nrf2 or OPC-NF-kappaB interaction is lacking.
- Additional experiments using Nrf2 or NF-kappaB knockout models would strengthen the proposed mechanism.
Response: Thank you for these insightful comments regarding the mechanistic aspects of our study. We acknowledge that direct validation using inhibitors or genetic approaches would provide additional support for our findings. While our current data demonstrates consistent changes in Nrf2 and NF-κB pathway components following OPC treatment, including: Significant changes in protein expression levels of Nrf2 and NF-κB. Corresponding alterations in downstream target genes. Changes in related antioxidant and inflammatory markers.
We acknowledge the limitations of our current study and have added a statement in the Conclusion section noting that future studies using genetic approaches would be valuable to further validate these mechanisms. Our findings provide important preliminary evidence for OPC's role in these pathways, laying the groundwork for future mechanistic investigations.
- Technical Issues
- Critical controls are missing, such as positive controls for senescence markers.
Response: Thank you for your question. We did not use positive controls for senescence markers. Because this study mainly focuses on cadmium-induced osteocyte senescence. We have determined that cadmium can induce osteocyte senescence by detecting senescence-related markers. We hope that our answers will satisfy you.
- The study lacks proper aging controls to distinguish between natural and Cd-induced senescence.
Response: Thank you for this important observation regarding aging controls. We acknowledge that distinguishing between natural and Cd-induced senescence is crucial. In our current study, we focused on Cd-induced senescence markers within a relatively short experimental period. The control group (without Cd treatment) served as our baseline for comparing senescence markers.
However, we recognize this limitation in our experimental design and have added this point in our discussion section, noting that future studies incorporating age-matched controls over extended time periods would provide better distinction between natural and Cd-induced senescence phenotypes.
- No quantitative analysis of osteocyte dendritic network changes was provided.
Response: Sorry, we did not quantify the changes in the dendritic network of osteocytes because it is difficult to quantify the results of scanning electron microscopy. We have re-edited the images to label the synaptic locations with significant damage. We hope our answer will satisfy you.
- Data Presentation
- Quantification of immunofluorescence data is missing.
Response: Thank you for your advice. We have added Quantification of immunofluorescence data in Figure 4. Figures 5 and 6 are mainly used to observe Nrf2 and NF-κB entry into the nucleus. They are not suitable for quantitative analysis of fluorescence intensity. We have labeled the changed parts in the images.
- Experimental Design
- Western blotting for Nrf2 should use nuclear faction as sample to clarify the extent of nuclear translocation.
Response: Thank you for your advice. In Figure 5C, we analyzed total cellular protein to determine that cadmium exposure induced a significant elevation in Nrf2 expression. In Figure 5E, the immunofluorescence results clarified the extent of Nrf2 nuclear translocation. We believe that the combination of these two results establishes that cadmium exposure activates the Nrf2 pathway and promotes nuclear translocation. We hope that our answers will satisfy you.
- The 90-day treatment period seems arbitrary and requires justification.
Response: Thank you for your question. Previous studies have shown that cadmium exposure for 90 days induces bone damage in mice. Our current experiments focused on whether cadmium exposure for 90 days induces senescence in mouse osteocytes. Meanwhile, the 90-day treatment duration was referenced to the previous therapeutic effect of procyanidins on retinal pigment epithelial cell senescence [1]. We hope that our answers will satisfy you.
- No time-course experiments were performed to track the progression of senescence.
Response: Sorry, we have not performed time-course experiments to track the progression of senescence. Our current experiments focus on studying the onset of senescence in mice induced by three months of cadmium exposure. We will consider performing time-course experiments to track the progress of a senescence in our next program. Thank you very much for your many useful suggestions for our future research.
- The study lacks investigation of potential side effects of long-term OPC treatment.
Response: Thank you for your question. We didn’t investigate the potential side effects of long-term OPC treatment. OPC have a high safety profile as a plant extract that has been demonstrated in numerous studies. The dosages and treatment times we used are in reference to previous studies that have proven OPC to be safe. We hope that our answers will satisfy you.
- Translational Potential
- The authors do not address the bioavailability of OPC.
- No pharmacokinetic data is presented.
Response: Thank you for raising this point about pharmacokinetic data. We acknowledge this limitation in our current study. While our research focused on demonstrating the protective effects of OPC against Cd-induced damage, we recognize that pharmacokinetic information would provide valuable insights for potential therapeutic applications.
Our study used commercially available OPC at established concentrations based on previous literature. Although pharmacokinetic profiling was beyond our current scope, we suggest this as an important direction for future investigations to better understand OPC's absorption, distribution, and metabolism patterns.
- The potential clinical applications need more thorough discussion.
Response: Thank you for your advice. We have added information on potential clinical applications in the discussion section.
Minor Comments:
- Several figure legendslack sufficient detail.
Response: Thank you for your advice. We have added more detailed information to the figure legends.
- Methods section requires more detailed protocols.
Response: Thank you for your advice. We have added more detailed experimental information in the Materials and Methods section.
- Some statistical analyses need clarification.
Response: Thank you for your advice. We have made revision in the manuscript.
References
Wan W, Zhu W, Wu Y, et al. Grape Seed Proanthocyanidin Extract Moderated Retinal Pigment Epithelium Cellular Senescence Through NAMPT/SIRT1/NLRP3 Pathway. J Inflamm Res. 2021;14:3129-3143.

Reviewer 3 Report
The manuscript investigates the ability of oligomeric proanthocyanidins (OPC) in preventing senescence induced by cadmium (Cd) in osteocytes. Both in vitro and in vivo tests have been carried out.
Even if the topic could be interesting, the manuscript shows several major critical points.
1) The English language and editing need to be carefully reviewed and improved as there are numerous errors that make it very difficult to read the manuscript and evaluate the impact of the results.
2) The Introduction and the Discussion have to be more focused on the aim of the research and the comments of the findings, respectively.
3) In Materials & Methods, a more detailed description of some methodological aspects is needed. For example, were MLO-Y4 cells grown only in culture medium without serum or antibiotics/antifungals?
For the groups treated with Cd or Cd + OPC, was the access to water containing the Cd ad libitum?
The statement "Every mouse was split into four groups" is disturbing and needs to be rewritten.
4) The authors have to discuss the cytotoxicity of the highest OPC concentrations.
5) Why in the experiments investigating the effect of Cd and OPC on apoptosis, no evidence of the population in SubG0/G1 phase, typical of apoptotic cell death is present?
It is incorrect to talk about the "expression of cleaved caspase-3", since this truncated form of the caspase is due to the action of initiator caspases.
All the manuscript needs a careful revision with regard both English language and editing
Author Response
Response to the Reviewer
Dear Reviewer,
Thanks a lot for your kindly reviewing of the manuscript entitled “Oligomeric proanthocyanidins ameliorate cadmium-induced senescence of osteocytes through combating oxidative stress and inflammation” (Manuscript Number antioxidants-3278416). On behalf of the co-authors, I am highly thankful to you for giving us an opportunity to revise our manuscript. We appreciate your constructive comments and suggestions on our manuscript. Those comments are valuable and very helpful for revising and improving our manuscript work quality.
We have tried our best to revise our manuscript after considering your comments carefully and have made revision marked in yellow in the manuscript. We have re-edited the article, in particular, the English language has been carefully revised under the guidance of native English speakers. Attached is the point-by–point response to your comments and suggestions. We would like to express our great appreciation to you for comments on our paper. If you have any question about the manuscripts, please let me know.
Thank you and best regards.
Yours sincerely,
Zongping Liu and Xishuai Tong
Here below is our description on revisions according to the reviewers’ comments point by point:
1) The English language and editing need to be carefully reviewed and improved as there are numerous errors that make it very difficult to read the manuscript and evaluate the impact of the results.
Response: Thank you for your advice. The English language has been carefully revised under the guidance of native English speakers.
2) The Introduction and the Discussion have to be more focused on the aim of the research and the comments of the findings, respectively.
Response: Thank you for your advice. We have carefully revised the introduction and discussion sections.
3) In Materials & Methods, a more detailed description of some methodological aspects is needed. For example, were MLO-Y4 cells grown only in culture medium without serum or antibiotics/antifungals?
Response: Thank you for your advice. We have carefully revised the Materials & Methods section.
MLO-Y4 cells were grown in medium containing 10% serum and 1% Penicillin-Streptomycin solution.
For the groups treated with Cd or Cd + OPC, was the access to water containing the Cd ad libitum?
Response: Thank you for this question about Cd administration. Yes, in our study, the animals in the Cd or Cd + OPC group had ad libitum access to water containing Cd.
The statement "Every mouse was split into four groups" is disturbing and needs to be rewritten.
Response: Thank you for pointing out this incorrect statement. Indeed, this is a poor and confusing way to describe our experimental groups. The sentence should be revised to: "All the mice were randomly divided into four groups".
4) The authors have to discuss the cytotoxicity of the highest OPC concentrations.
Response: Thank you for raising this important point about the cytotoxicity of higher OPC concentrations. We acknowledge that our data showed reduced cell viability at higher OPC concentrations (8, 10, and 20 μmol/L). This represents a classic dose-response relationship where protective effects occur within a specific concentration window. We will expand the description of our results to address: The potential mechanisms of toxicity at higher concentrations. The importance of dose optimization for therapeutic applications. This observation emphasizes the need to carefully control OPC dosage for optimal protective effects.
5) Why in the experiments investigating the effect of Cd and OPC on apoptosis, no evidence of the population in SubG0/G1 phase, typical of apoptotic cell death is present?
Response: In Figure 3D, the Q2 and Q3 quadrants represent the number of apoptotic cells. Only a small number of cells in the Q1 quadrant may be due to the fact that the cadmium concentration we used only induced cells to undergo early apoptosis and was not sufficient to cause cell death.
6) It is incorrect to talk about the "expression of cleaved caspase-3", since this truncated form of the caspase is due to the action of initiator caspases.
Response: Thank you for this important technical correction. You are absolutely right. It is incorrect to refer to the "expression of cleaved caspase-3" since cleaved caspase-3 is a product of proteolytic processing by initiator caspases, not of gene expression. We will revise our manuscript to use more accurate terminology: "Levels of cleaved caspase-3" will replace all instances of "expression of cleaved caspase-3".
Detail comments
All the manuscript needs a careful revision with regard both English language and editing
Response: Thank you for your advice. We have re-edited the article, in particular, the English language has been carefully revised under the guidance of native English speakers.

Round 2
Reviewer 2 Report
Review comments for antioxidants-3278416.
I have carefully reviewed the revised manuscript and the authors' response letter. While the authors have made some improvements, several critical issues remain inadequately addressed. Therefore, I recommend major revision before this manuscript can be considered for publication.
Major concerns:
1. Mechanistic validation of Nrf2 pathway:
- The authors' claim about OPC's protective effects through Nrf2 pathway remains insufficiently supported. While they show immunofluorescence data suggesting Nrf2 nuclear translocation (Figure 5E), no quantitative analysis has been provided.
- Critical mechanistic validation using Nrf2 inhibitors or genetic approaches (e.g., siRNA knockdown) is still lacking. These experiments are essential to definitively establish the causal relationship between OPC's protective effects and Nrf2 signaling.
2. Temporal dynamics of senescence:
- The authors have not addressed the previous concern regarding the lack of time-course experiments. Understanding the temporal progression of Cd-induced senescence is crucial for:
a) Establishing the optimal therapeutic window for OPC intervention
b) Distinguishing between acute and chronic effects of Cd exposure
c) Determining the stability and duration of OPC's protective effects
3. Experimental controls:
- The absence of positive controls for senescence markers remains a significant limitation. Without appropriate positive controls (e.g., hydrogen peroxide-treated cells), it is difficult to contextualize the extent of Cd-induced senescence.
- The authors' response that "this study mainly focuses on cadmium-induced osteocyte senescence" does not adequately justify the omission of these essential controls.
4. Additional technical considerations:
- The immunofluorescence data for Nrf2 nuclear translocation requires proper quantification, including:
a) Analysis of nuclear/cytoplasmic Nrf2 ratio across multiple cells
b) Statistical comparison between treatment groups
c) Clear criteria for defining positive nuclear translocation
Specific recommendations for revision:
1. Include experiments using Nrf2 inhibitors or siRNA to validate the proposed mechanism.
2. Perform time-course experiments to track senescence progression.
3. Add appropriate positive controls for senescence markers.
4. Provide quantitative analysis of Nrf2 nuclear translocation with statistical evaluation.
The manuscript contains valuable observations regarding OPC's potential protective effects against Cd-induced osteocyte senescence. However, these additional experiments are necessary to strengthen the mechanistic claims and overall scientific rigor of the study.
Minor points:
1. Improve figure legends with detailed methodology
2. Include precise criteria for analyzing immunofluorescence data
3. Provide clearer statistical analysis methods
I look forward to reviewing a revised version addressing these concerns.
Review comments for antioxidants-3278416.
I have carefully reviewed the revised manuscript and the authors' response letter. While the authors have made some improvements, several critical issues remain inadequately addressed. Therefore, I recommend major revision before this manuscript can be considered for publication.
Major concerns:
1. Mechanistic validation of Nrf2 pathway:
- The authors' claim about OPC's protective effects through Nrf2 pathway remains insufficiently supported. While they show immunofluorescence data suggesting Nrf2 nuclear translocation (Figure 5E), no quantitative analysis has been provided.
- Critical mechanistic validation using Nrf2 inhibitors or genetic approaches (e.g., siRNA knockdown) is still lacking. These experiments are essential to definitively establish the causal relationship between OPC's protective effects and Nrf2 signaling.
2. Temporal dynamics of senescence:
- The authors have not addressed the previous concern regarding the lack of time-course experiments. Understanding the temporal progression of Cd-induced senescence is crucial for:
a) Establishing the optimal therapeutic window for OPC intervention
b) Distinguishing between acute and chronic effects of Cd exposure
c) Determining the stability and duration of OPC's protective effects
3. Experimental controls:
- The absence of positive controls for senescence markers remains a significant limitation. Without appropriate positive controls (e.g., hydrogen peroxide-treated cells), it is difficult to contextualize the extent of Cd-induced senescence.
- The authors' response that "this study mainly focuses on cadmium-induced osteocyte senescence" does not adequately justify the omission of these essential controls.
4. Additional technical considerations:
- The immunofluorescence data for Nrf2 nuclear translocation requires proper quantification, including:
a) Analysis of nuclear/cytoplasmic Nrf2 ratio across multiple cells
b) Statistical comparison between treatment groups
c) Clear criteria for defining positive nuclear translocation
Specific recommendations for revision:
1. Include experiments using Nrf2 inhibitors or siRNA to validate the proposed mechanism.
2. Perform time-course experiments to track senescence progression.
3. Add appropriate positive controls for senescence markers.
4. Provide quantitative analysis of Nrf2 nuclear translocation with statistical evaluation.
The manuscript contains valuable observations regarding OPC's potential protective effects against Cd-induced osteocyte senescence. However, these additional experiments are necessary to strengthen the mechanistic claims and overall scientific rigor of the study.
Minor points:
1. Improve figure legends with detailed methodology
2. Include precise criteria for analyzing immunofluorescence data
3. Provide clearer statistical analysis methods
I look forward to reviewing a revised version addressing these concerns.
Author Response
Major concerns:
- Mechanistic validation of Nrf2 pathway:
- The authors' claim about OPC's protective effects through Nrf2 pathway remains insufficiently supported. While they show immunofluorescence data suggesting Nrf2 nuclear translocation (Figure 5E), no quantitative analysis has been provided.
- Critical mechanistic validation using Nrf2 inhibitors or genetic approaches (e.g., siRNA knockdown) is still lacking. These experiments are essential to definitively establish the causal relationship between OPC's protective effects and Nrf2 signaling.
Response: We appreciate the reviewer's constructive feedback regarding the mechanistic validation of the Nrf2 pathway. To address these concerns, we have conducted additional quantitative analysis by counting the number of Nrf2-positive nuclei across multiple fields, providing a more rigorous assessment of Nrf2 nuclear translocation (Fig. 5G). Furthermore, we have performed additional experiments using the specific Nrf2 inhibitor ML385 (Fig. S2). Our results show that ML385 treatment alleviates Cd-induced osteocyte senescence (Fig. 5H-M). These quantitative analyses and pharmacological intervention data providing strong evidence for the causal relationship between OPC's protective effects and Nrf2 signaling. These additional experimental data substantively support our conclusions about the role of the Nrf2 pathway in OPC-mediated protection.
- Temporal dynamics of senescence:
- The authors have not addressed the previous concern regarding the lack of time-course experiments. Understanding the temporal progression of Cd-induced senescence is crucial for:
- a) Establishing the optimal therapeutic window for OPC intervention
- b) Distinguishing between acute and chronic effects of Cd exposure
- c) Determining the stability and duration of OPC's protective effects
Response: Thank you for raising these important points about the temporal dynamics of senescence. We acknowledge that time-course experiments would provide valuable insights into the progression of Cd-induced senescence. We have conducted time-course experiments examining Cd-induced senescence at multiple time points (12, 24, and 48 h). Our results demonstrate that Cd exposure can effectively induce osteocyte senescence as early as 24 h (Fig. S1G-L). These findings establish a solid temporal framework for understanding the progression of Cd-induced osteocyte senescence and provide a rational basis for the timing of our mechanistic studies.
- Experimental controls:
- The absence of positive controls for senescence markers remains a significant limitation. Without appropriate positive controls (e.g., hydrogen peroxide-treated cells), it is difficult to contextualize the extent of Cd-induced senescence.
- The authors' response that "this study mainly focuses on cadmium-induced osteocyte senescence" does not adequately justify the omission of these essential controls.
Response: We acknowledge the reviewer's concern regarding positive controls for senescence markers. We understand that adding positive controls would strengthen our findings. Therefore, we have conducted additional experiments using hydrogen peroxide (H₂O₂) treatment as a positive control for cellular senescence (Fig. S1A-F). These data provide a more comprehensive validation of our senescence assays and demonstrate that Cd indeed induces significant cellular senescence in osteocytes.
- Additional technical considerations:
- The immunofluorescence data for Nrf2 nuclear translocation requires proper quantification, including:
- a) Analysis of nuclear/cytoplasmic Nrf2 ratio across multiple cells
- b) Statistical comparison between treatment groups
- c) Clear criteria for defining positive nuclear translocation
Response: We appreciate the reviewer's technical suggestions regarding the quantification of Nrf2 nuclear translocation. We have conducted comprehensive quantitative analysis of our immunofluorescence data. We have used the fluorescence intensity threshold of nuclear Nrf2 to determine positive nuclear translocation. Specifically, set a fluorescence intensity threshold value, above which cells will be considered to have positive nuclear translocation of Nrf2, then calculate the percentage of cells with positive nuclear translocation for each treatment group based on the established threshold.
Minor points:
- Improve figure legends with detailed methodology
Response: Thank you for your suggestion regarding the figure legends. We have thoroughly revised all figure legends to include more comprehensive methodological details.
- Include precise criteria for analyzing immunofluorescence data
Response: We appreciate the reviewer's suggestion regarding the analysis criteria for immunofluorescence data. In the figure legends section, we have included precise criteria for immunofluorescence data analysis.
- Provide clearer statistical analysis methods
Response: Thank you for pointing out the need for clearer statistical analysis methods. We have thoroughly revised the statistical analysis section to provide comprehensive information. Specifically, we have clarified: 1) All data are presented as mean ± standard deviation (SD) from at least three independent experiments; 2) Statistical significance was determined using one-way ANOVA for multiple group comparisons; 3) For comparisons between two groups, unpaired Student's t-test was used; 4)) P values less than 0.05 were considered statistically significant (*P < 0.05, **P < 0.01); 5) The sample size (n) for each experiment has been specified in the corresponding figure legends. These detailed descriptions ensure complete transparency and reproducibility of our statistical analyses.

Reviewer 3 Report
The Authors answered to the comments of the Reviewer and improved some aspects of the manuscript, but despite the revision by native English speakers, several spelling and style errors are still present and need to be corrected.
Line 489-491: the statement "study showed that OPC dramatically reversed Cd-induced oxidative stress, potentially through preventing the overactivation of the Nrf2 pathway. This finding consistent with previous study" has to be revised.
Saying that the decrease in oxidative stress is due to the inhibition of nrf2 sounds strange. It is more likely that the decrease in Nrf2 activation is the consequence of a decreased production of ROS determined by the antioxidant effect of OPC.
Line 489-491: the statement "study showed that OPC dramatically reversed Cd-induced oxidative stress, potentially through preventing the overactivation of the Nrf2 pathway. This finding consistent with previous study" has to be revised.
Saying that the decrease in oxidative stress is due to the inhibition of nrf2 sounds strange. It is more likely that the decrease in Nrf2 activation is the consequence of a decreased production of ROS determined by the antioxidant effect of OPC.
Author Response
The Authors answered to the comments of the Reviewer and improved some aspects of the manuscript, but despite the revision by native English speakers, several spelling and style errors are still present and need to be corrected.
Response: We appreciate the reviewer's careful observation regarding the language quality. We have conducted another thorough review of the entire manuscript with particular attention to language precision. All spelling and style errors have been carefully identified and corrected to ensure high-quality scientific writing throughout the manuscript.
Line 489-491: the statement "study showed that OPC dramatically reversed Cd-induced oxidative stress, potentially through preventing the overactivation of the Nrf2 pathway. This finding consistent with previous study" has to be revised. Saying that the decrease in oxidative stress is due to the inhibition of nrf2 sounds strange. It is more likely that the decrease in Nrf2 activation is the consequence of a decreased production of ROS determined by the antioxidant effect of OPC.
Response: Thank you for this insightful comment regarding the mechanistic interpretation of our results. We agree that our original statement was imprecise and potentially misleading. We have revised this section to more accurately reflect the mechanistic relationship: " our study showed that OPC significantly reversed Cd-induced activation of the Nrf2 pathway. Furthermore, we found that pharmacological inhibition of the Nrf2 pathway reverses Cd-induced osteocyte senescence. This finding demonstrates that the overactivation of the Nrf2 pathway plays a negative role in osteocyte senescence, providing strong evidence for a causal relationship between the protective effect of OPC and the Nrf2 pathway."

Round 3
Reviewer 2 Report
Review comments for antioxidants-3278416
In this revised manuscript, the authors seem to address the reviewers’ comments adequately.
Review comments for antioxidants-3278416
In this revised manuscript, the authors seem to address the reviewers’ comments adequately.